# The Efficiency of a Low Dose of Biochar in Enhancing the Aromaticity of Humic-Like Substance Extracted from Poultry Manure Compost

**Keiji Jindo [1,2,\*], Miguel A. Sánchez-Monedero [2]**  **, Kazuhiro Matsumoto [3]**  **and Tomonori Sonoki [4]** 

1   Plant Production System Group, Wageningen University and Research,
    6708PB Wageningen, The Netherlands
2   Department of Soil and Water Conservation and Organic Waste Management, CEBAS-CSIC, P.O. Box 4195,
    30080 Murcia, Spain; monedero@cebas.csic.es
3   Faculty of Agriculture, Shizuoka University, Shizuoka 422-8529, Japan; matsumoto.kazuhiro@shizuoka.ac.jp
4   Faculty of Agriculture and Life Science, Hirosaki University, Hirosaki 036-8561, Japan;
    sonoki@hirosaki-u.ac.jp
\*   Correspondence: keijindo@gmail.com; Tel.: +31-(0)-317-482141

**Abstract:** Using biochar as a bulking agent in composting is gradually becoming popular for the minimization of nitrogen losses during the process and the improvement in compost quality. While a wide range of different biochar doses is applied, not much clear information was available about the optimum ratio. This study presents the impact of adding a low dose (2% *v/v*) of slow-pyrolysis oak biochar (*Quercus serrate* Murray), into poultry manure on the recalcitrant characteristic of humified organic matter. The influence in the chemical composition of humic-like substance was evaluated in poultry manure compost prepared with (PM+B) and without biochar (PM). The shift to slightly more stable chemical composition was shown in humic acid-like (HA) and fulvic acid-like (FA) extracted from PM+B compost, by increasing the proportion of aromatic carbon groups and thermal stability measured by thermogravimetry. We conclude that the addition of 2% biochar moderately enhances the recalcitrance of humified organic carbon and this could be feasible for the implementation of the biochar use in composting since only a small amount is required.

**Keywords:** biochar; humification; NMR; pyrolysis; composting

## 1. Introduction

Producing bioenergy from biomass pyrolysis provides by-products as the recalcitrant char material, now termed "biochar", which can sequester carbon by maintaining long half-life in soil. In addition, other positive effects of biochar application have been observed, such as improvement of soil quality and retention of pollutant compounds (pesticides, heavy metals, and volatile organic compounds) [1–3]. In particular, the influence of biochar on crop growth has been considered to be a promising tool to shed light on the tackling against global food-security to boost up the production [4]. Impact of biochar application on crop growth increases remarkably when used in combination with other components such as mineral fertilizers, earthworms and composts [5–8]. Compost material is a useful option to use biochar as an additive for enhancing the composting process by reducing the nitrogen losses and improving the final product [9–12]. Several authors have proposed a synergistic effect between biochar and the composting process [13,14]. However, it remains necessary to research the complex mechanism of interaction between composting materials and biochar during the biodegradation process. Several mechanisms have been proposed for the participation of biochar into the humification

of organic matter during composting, including (1) the abiotic decomposition of biochar, (2) adsorption of easy-degradable compounds and fulvic acids in biochar surface, (3) and favouring the activity of specific microorganisms involved in the degradation of humic substances [15]. Furthermore, given that a wide-range of application rates, ranging from 2% to 50% in volume base, has been applied for composting process [8,16–19], identification of an optimum proportion of biochar addition would be necessary for decision-making on the future use of biochar [20,21]. Other reports proved an increase in the content of humified organic carbon by biochar addition in the range between 13–35% and 15–42%, for fulvic-like and humic-like acids, respectively [17,22], and they conclude that high biochar doses (>12%) are needed to create an impact on the humification. To make biochar use easily feasible and applicable for composting, the profitability needed to be considered counting on the cost of feedstock purchase, transportation, biochar production, maintenance, labor and storage [20,23–25]. Furthermore, the economic and environmental trade-off with other feedstock uses such as biogas and biofuel has to be taken into account [26,27]. Hence, it is worthwhile to evaluate the lowest dose biochar that can cause an improvement of the humification during the composting process.

In our previous study, we presented the chemical and biochemical assessment of the composting process of poultry manure with biochar (2% *v/v*) of the hard-wood tree (*Quercus serrate* Murray) as a bulking agent [28]. The addition of such small dose of biochar was effective in changing the chemical properties of the bulk OM and also in the enhancement in the aromaticity indices. Furthermore, a strong change was observed in enzymatic activities as well as the content of dissolved organic carbon, reflecting the interaction of biochar on the composting material. Although the compost maturity with biochar addition was evaluated by the quantification of alkaline-extractable carbon as humified organic carbon, showing 10% higher content than non-biochar addition compost [25], further research on the chemical composition of the extracted humic-like substances was not explored.

As a continuity study, this work presents the effect of a small dose of biochar addition (2% in volume) on chemical and structural characteristics of the extracted humic-like acid (HA) and fulvic-like acid (FA) by using Nuclear Magnetic Resonance (NMR), Fourier-transform infrared spectroscopy (FT-IR) and Thermogravimetry (TGS).

## 2. Materials and Methods

### 2.1. Pyrolysis Process

In our study, hard-wood tree (*Quercus serrate* Murray) was used as feedstock. The pyrolysis process was conducted by Tree-Work Ltd. (Aomori, Japan). Biochar production by slow pyrolysis was carried out using a traditional Japanese charcoal kiln during 24-48 h at atmospheric pressure maintaining the temperature in a range between 400 and 600 °C. The main physicochemical biochar properties were: pH ($H_2O$) = 7.23; C = 791.5 g $kg^{-1}$; O = 91.5 g $kg^{-1}$; N = 37.6 g $kg^{-1}$; H = 18.9 g $kg^{-1}$; ash = 78.7 g $kg^{-1}$, methylene blue adsorption capacity (8.3 mg $g^{-1}$), Surface area = 255.0 $m^2$ $g^{-1}$. The elemental contents were analysed by automatic elemental analysis (LECO CHNS-932, Saint Joseph, MI, USA). Ash content was conducted according to the American Society for Testing and Materials (ASTM) D1752-84. The pH was measured on a 1:10 (*w/v*) water extract with the MP220 pH meter. The specific surface area was determined using $N_2$ sorption isotherms. More details about the description of pyrolysis and biochar properties can be found elsewhere [28].

### 2.2. Composting Process

Two composting piles (PM and PM+B) were set up by mixing poultry manure (20% of the total amount in volume base) with apple pomace (50%), rice husk (20%) and oak bark was used (10%) as a bulking agent. Mixture PM + B was prepared by enriching the PM mixture with a 2% addition of biochar (in volume). The size of the composting pile was about 3 tons with a cone shape windrow of approximately 2-m height. The piles were turned twice a month to aerate and homogenize the composting materials. The composting process lasted approximately 150 days for both PM and

PM + B. A full description of the composting process and the monitoring method by temperature, polymerization, and C: N ratios were described in the previous study [28]. A representative sample was obtained after turning (to ensure homogenization) by mixing different subsamples randomly collected from at least six different locations in the pile after maturation (150 days of composting). These samples were collected, air-dried and ground to 0.5 mm. Main properties of PM and PM+B are shown in Table 1.

**Table 1.** Selected chemical-physical properties of poultry manure (PM) and poultry manure blended compost with biochar (PM+B).

| | EC (dSm$^{-1}$) | pH (%) | C (%) | C/N | Alkali-extractable Carbon (g kg$^{-1}$) |
|---|---|---|---|---|---|
| PM | 3.8 (0.1) *[1] | 7.3 (0.1) | 32.7 (0.3) | 17.8 (0.5) | 22.9 (0.3) |
| PM+B | 3.9 (0.1) | 7.6 (0.1) | 36.6 (0.1) | 21.7 (0.2) | 25.3 (0.3) |

*[1] Standard deviation in brackets ($n = 3$).

### 2.3. Extraction of HA and FA from the Composted Materials (PM and PM+B).

A full description of the extraction method is written in a previous study [29]. In short, the extraction of the humic substances was carried out with 10 g of mature compost samples and 100 mL of 0.25 M NaOH, under an atmosphere of $N_2$. The extracts were collected and centrifuged. The filtered solutions were acidified with $H_2SO_4$ to pH 2 and kept for 24 h at 4 °C; they were then centrifuged to separate the precipitated HA from the supernatant FAs. The purification of FAs was done with XAD-8 resin (No. 20,278 Supelco Supelite$^{TM}$, Sigma Aldrich, (St. Louis, MO, United States). The adsorbed FAs were then eluted using one-bed volume of 0.1 M NaOH. The Na-fulvates were then passed through a strongly acidic cation exchange resin (No. 216,534 Aldrich Amberlite IR 120+ hydrogen form) to obtain saturated H+ FA. Finally, the FA samples were freeze-dried to keep the material stable until use. The HAs were purified by washing with a dilute HF–HCl solution. This procedure was repeated three times. After centrifugation, the sample was washed repeatedly with water, followed by dialysis against deionized water. The dialyzate was freeze-dried for chemical characterization.

### 2.4. HA and FA Characterization: Elemental Composition, NMR, FT-IR, Thermogravimetry

The elemental composition of C, H, and N was determined using an elemental analyzer (Thermo Finnigan EA-1112, Thermo Fisher Scientific Inc., Waltham, MA, USA) and calculated on ash-free basis; the O content was calculated by difference (i.e., O(%) = 100 − C(%) − H(%)− N(%))and can therefore include trace fractions of S and/or P. FT-IR spectroscopy was analysed on a Varian 670-IR (Agilent Technologies Inc., Santa Clara, CA, USA) using the pellet technique by mixing 1 mg of dried biochar with 300 mg of pre-dried and pulverized spectroscopic-grade KBr from Merck and Co., Whitehouse Station, NJ, USA. Cross-polarization magic angle spinning (CPMAS) $^{13}$C nuclear magnetic resonance ($^{13}$C-NMR) spectra were acquired from the solid samples with a Varian 300, equipped with a 4 mm wide bore MAS probe, operating at a $^{13}$C resonating frequency of 75.47 MHz. The spectra were integrated in the chemical shift (ppm) resonance intervals as follow [26]: alkyl C (0–45 ppm), N-alkyl/methoxyl C (45–65 ppm), O-alkyl C (65–90 ppm), anomeric C (90-108 ppm), aromatic C and phenol C (108–160 ppm), and carbonyl C (160–210 ppm). The degree of aromaticity (%) was calculated as following [30]: aromatic C × 100/(alkyl C + N-alkyl C + O-alkyl C + aromatic C). Thermal analysis of biochar materials was measured using an SDT-2960 simultaneous differential scanning calorimeter–thermal gravimetric analyzer (TA Instruments, New Castle, DE). Thermal analyses were conducted under a static-air atmosphere as follows: a temperature equilibrating at 30 °C followed by a linear heating rate of 5 °C min$^{-1}$ from 30 to 105 °C at which point, an isotherm was maintained during 10 min, and then ramping continued at 5 °C min$^{-1}$ from 105 to 680 °C. The ash content was calculated from the inorganic residue remaining at the end of the ramp. The main weight losses occurred in the 110 to 350 °C ($W_1$) and 350 to 550 °C ($W_2$) ranges. In addition, the ratio $W_2/W_1$ was deployed as a thermal lability index of the organic materials [31].

*2.5. Statistic Analysis*

Data of elemental analysis of the humic and fulvic acids were submitted to statistical analysis by the *t*-student test. For the statistical analyses, Rstudio program (3.3 version, RStudio Inc., Boston, MA, USA) was used.

## 3. Results and Discussion

*3.1. Elemental Analysis and Thermal Stability of the Extracted Humic Fractions*

Table 2 shows the elemental composition and the thermal stability analysis of the humified organic carbon (HA and FA) extracted from PM and PM+B. The result obtained in our study generally aligns with those extracted from soils [32], and the difference was shown in a lower proportion of C and a higher proportion of other elements in our study, compared to those from soils. This could be due to the different time-span of the humification that determines the different characteristics between soil humic substance and humic-like substance from compost. In our study, HA and FA extracted from PM had slightly higher C concentration than those extracted from PM+B. However, the HA extracted from PM+B had lower H:C ratio than that from PM, reflecting the higher levels of aromatic compounds formed by composting and illustrates a higher degree of humification [33–35]. This is in accordance with the result of $W_2/W_1$ ratio, known as thermal stability indicator of humic structures [29], showing 38% and 40% higher in HA and FA of PM+B, respectively, compared to those of PM. These results imply that more stable chemical structure was formed with the biochar presence [36]. Biochar produced from wood material contains a high abundancy of oxygen functional groups on its surfaces, which enhances the biochar capacity for the absorption of nutrients, moisture and dissolved organic matter. Consequently, the absorbed compounds which are retained in the biochar surface may participate in chemical and biochemical reactions associated with humification [17]. While H:C is a well-known indicator of aromaticity, O:C represents the presence of carboxylic and carbohydrate carbon [28], which is more dominant in FA with confirmation by higher O:C than HA (Table 2). Those ratios found in this study are between typical range in HA and FA extracted from different compositing mixtures as reported by [36–38].

**Table 2.** The elemental composition and thermogravimetric ratio of humic acid-like (HA) and fulvic acid-like (FA) extracted from poultry manure compost (PM) and poultry manure blended with biochar (PM+B).

| | HA | | | | | | | FA | | | | | | |
|---|---|---|---|---|---|---|---|---|---|---|---|---|---|---|
| **Origin** | **Mass/% ash-free basis** | | | | **H:C** | **O:C** | **$W_2/W_1$** [*1] | **Mass/% ash-free basis** | | | | **H:C** | **O:C** | **$W_2/W_1$** |
| | **C** | **H** | **N** | **O** | | | | **C** | **H** | **N** | **O** | | | |
| PM | 55.1 | 8.0 | 6.1 | 30.7 | 1.7 | 0.4 | 0.71 | 38.6 | 4.9 | 5.8 | 50.6 | 1.5 | 0.9 | 0.49 |
| PM+B | 54.0 | 6.1 | 7.6 | 32.2 | 1.4 | 0.5 | 0.98 | 37.6 | 5.7 | 6.1 | 50.9 | 1.8 | 1.0 | 0.69 |
| *SE* [*2] | *0.09* | *0.10* | *0.03* | *0.14* | *0.02* | *0.00* | - | *0.19* | *0.20* | *003* | *0.01* | *0.07* | *0.01* | - |
| *Significance* [*3] | ** | ** | ** | ** | ** | ** | - | * | NS | * | NS | . | . | - |

[*1] Ratio between the mass losses associated with the second ($W_2$) and first ($W_1$) exothermic reactions of thermal analysis; [*2] SE, pooled standard error (*n* = 2) of the chemical analysis measured according to [39] [*3] For each column, means were subjected to statistical analysis by the t-student test. NS, ., *, and ** indicate non-significant and significant differences at the 0.1, 0.05 and 0.01 levels respectively.

*3.2. NMR Characterisation of HA and FA Extracted from Mature Compost*

Figure 1 and Table 3 represent the quantitative analysis of the chemical composition of HA and FA obtained by CPMAS $^{13}$C-NMR. This technic allows quantifying the proportions of different carbon groups of organic matter [30]. Similar trends of $^{13}$C-NMR (Figure 1) are shown in both HAs and FAs extracted from PM and PM+B, characterised by the presence of peaks at 56 and 72 ppm of alkyl C, at 130 ppm of aromatic C, and at 175 ppm of carboxylic C. A difference between HAs and FAs is observed in the range of carboxyl and carbonyl C (160–185 and 185–225 ppm) in Table 3, where FAs have larger

proportion of these carbon groups than HAs, and this feature is a typical characteristic of FA [40]. In Figure 1, PM exhibits shaper peaks in the alkyl C range in HA and FA, while PM+B demonstrates larger peak intensity in the aromatic C group. This difference was reflected quantitatively by the proportion of the aromatic C group (Table 3). Also, the difference between PM+B and PM was found in the presence of the moderate peak of carbonyl (C = O) at around 190 ppm in HA and FA. In particular, this was more clearly shown in HA than in FA (Figure 1).

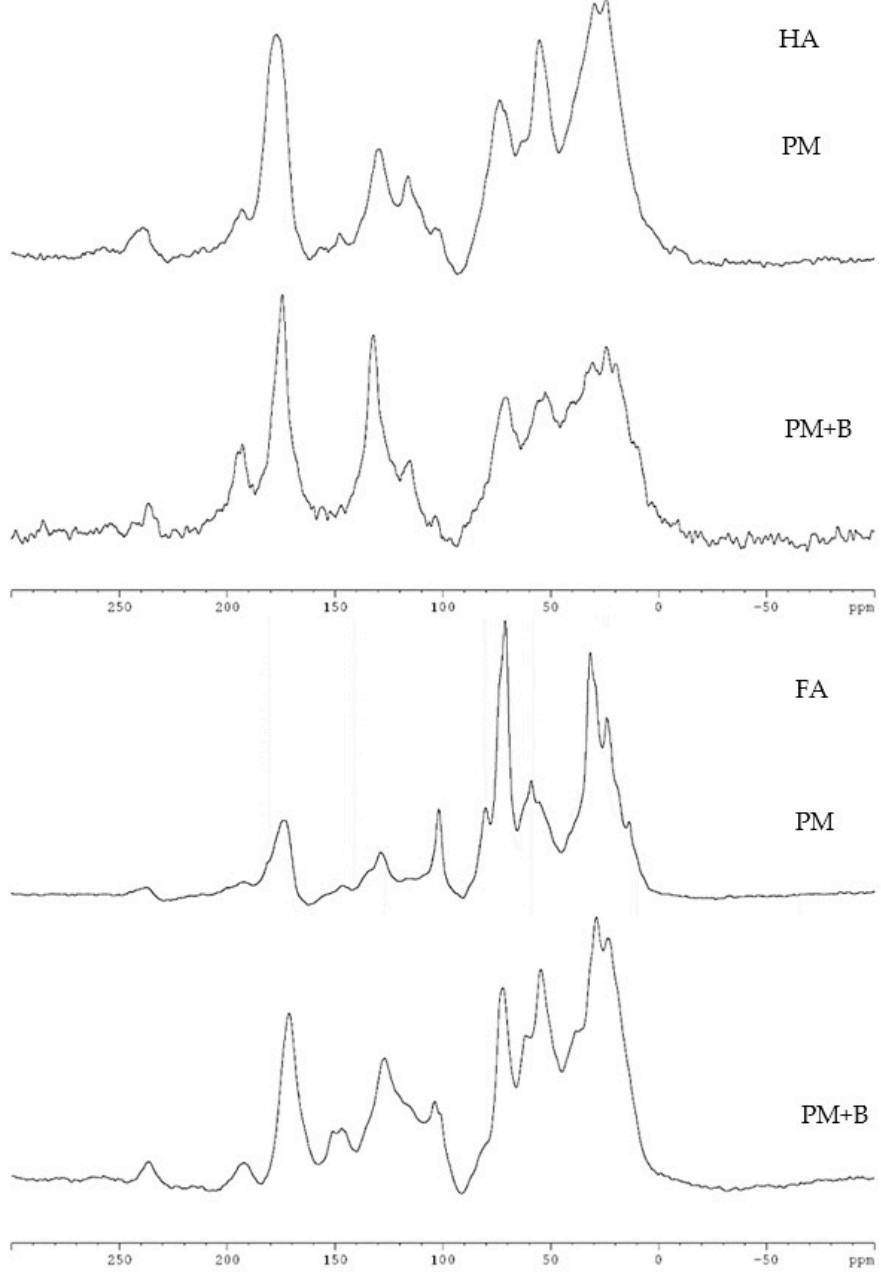

**Figure 1.** Cross-polarization magic angle spinning $^{13}$C nuclear magnetic resonance (CPMAS $^{13}$C-NMR) of humic acid-like (HA) and fulvic acid-like (FA) extracted from poultry manure compost (PM) and poultry manure blended with biochar (PM+B).

Aromaticity and hydrophobicity of humified organic matter play an important role in reinforcing the recalcitrant characteristics against the degradation, and these indicators are considered to be the composting maturity [41,42]. Table 4 shows the aromaticity and hydrophobicity of humified organic carbon (HA and FA) in our study. Higher aromaticity index is shown in HA extracted from PM+B,

almost doubling that from PM due to the increased proportion in aromatic C (108–160 ppm), shown in Table 2. FA has a similar result but less impact. Compared to another report on the effect of a high-dose biochar addition (10% *v/v*) into manure compost [43], the low-dose addition (2% *v/v*) in our study has the same range of aromaticity and hydrophobicity in both HA and FA.

**Table 3.** Relative abundances of different carbon (in %), measured by [13]C CPMAS NMR of humic acid-like (HA) and fulvic acid-like (FA) in maturation phase extracted from poultry manure compost (PM) and poultry manure blended with biochar (PM+B).

| Origin | ppm | | | | | | |
|---|---|---|---|---|---|---|---|
| | 0–45 | 45–65 | 65–95 | 95–108 | 108–160 | 160–185 | 185–225 |
| *HA* | | | | | | | |
| PM | 34.7 | 14.3 | 22.2 | 4.4 | 10.3 | 7.8 | 4.4 |
| PM+B | 34.1 | 16.4 | 11.5 | 4.3 | 19.9 | 10.4 | 3.3 |
| *FA* | | | | | | | |
| PM | 34.3 | 15.4 | 12.1 | 2.0 | 15.0 | 14.4 | 6.8 |
| PM+B | 31.3 | 13.2 | 11.1 | 1.0 | 20.0 | 14.7 | 8.5 |

**Table 4.** Aromatic Index and Hydrophobicity (HB/HI) of humic acid-like (HA) and fulvic acid-like (FA) extracted from poultry manure compost (PM) and poultry manure blended with biochar (PM+B).

| Origin | HA | | FA | |
|---|---|---|---|---|
| | Aromatic Index [*1] | HB/HI [*2] | Aromaticity Index | HB/HI |
| PM | 12.6 | 0.9 | 19.5 | 1.1 |
| PM+B | 24.3 | 1.3 | 26.4 | 1.3 |

[*1] Aromatic Index = (aromatic C + phenolic C)/(alkyl C + N-alkyl C + O-alkyl C + aromatic C + phenolic C) × 100.
[*2] Hydrophobic C/Hydrophilic C (HB/HI) = (alkyl C + aromatic C+ phenolic C)/(N-alkyl C + O-alkyl C + carboxyl, amides, ester).

### 3.3. FT-IR Characterization of HA and FA from Mature Compost

Figure 2 represents the result of FT-IR in HAs and FAs extracted from PM and PM+B. Intense bands of aliphatic structures C-H (2920 cm$^{-1}$ and 2855 cm$^{-1}$) appear in the both of HA and FA of PM while those peaks appear with more moderate shape in PM+B. The broad around 1060 cm$^{-1}$ attributed to aliphatic C-O stretching [44] is clearly shown in HA of PM+B while the decline in the peak intensities of aromatic C = C ring stretching (1577 cm$^{-1}$) was notably observed in HA of PM. The high-dose additions (>8%) can contribute to the humification process during composting, as observed in [22,45,46]. This study demonstrated that even a small amount of the biochar addition (2%) could reinforce the aromaticity of the HA and FA which is an important characteristic that provides recalcitrance against the degradation [42].

It should be noted that the extraction method of the humification is complementary to the analysis of the bulk organic matter, and it can cause a bias to the analysis of the chemical composition of the humic substance [47]. To consider this matter, monitoring the chemical and physical property of bulk compost [48] together the segmented study of humified organic matter would be a holistic approach. However, the extraction method has the advantage to overcome the limitation of the presence of small particles of biochars in composting that can be hardly removed from the compost matrix before the analysis.

A couple of reports on the application of the different biochar doses for the composting have been published [18,22,45,46,49], and all of them observed a better impact of biochar on humification when biochar was used at high doses, compared to low-doses. In addition, this result could be attributed to the limited surface area on biochar for absorption in low-dose application [49,50]. However, a small dose of biochar has shown to positively affect the composting process of poultry manure [51], and

this impact was due to the enhancement of the environmental conditions for microbial growth in the composting pile that can also affect OM degradation and humification.

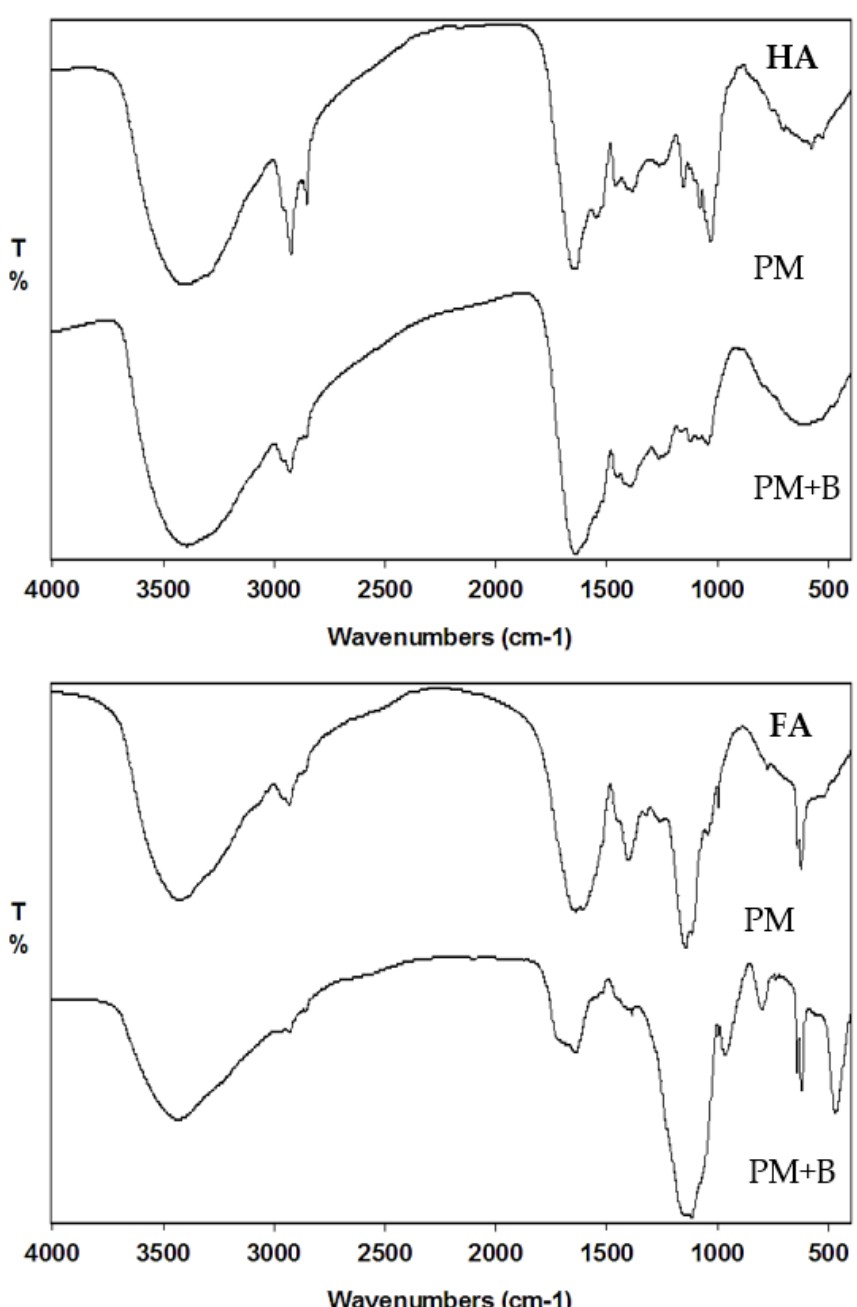

**Figure 2.** Fourier transformed infrared (FT-IR) spectroscopy of humic acid-like (HA) and fulvic acid-like (FA) extracted from poultry manure compost (PM) and poultry manure blended with biochar (PM+B).

## 4. Conclusions

The low biochar addition (2% in volume) into the composting mixture contributed to creating the more stable chemical structure of the HA and FA by increasing 38% and 40% of $W_2/W_1$ in HA and FA, respectively. The reinforcement of recalcitrant characteristic was reflected by lower H:C ratio, higher aromaticity, and hydrophobicity measured by elemental analysis and spectra techniques (FT-IR and NMR). Until now, no much information was available for the biochar influence on the chemical composition of humified organic carbon with a small dose of the biochar addition into composting. Our study can demonstrate that low-dose application is beneficial to improve the final compost product

and minimize the cost of the biochar production. Despite these findings, further research is needed for a better understanding of the interaction of biochar with the humification process during composting such as biochar abiotic and biotic degradation and its effect on humification.

**Author Contributions:** "conceptualization, K.J. and M.A.S.-M.; methodology M.A.S.-M.; software, K.J.; validation, K.J., M.A.S.-M. and T.S.; formal analysis, K.J., M.A.S.-M., K.M. and T.S.; investigation, K.J. and T.S.; resources, T.S. and M.A.S.-M.; data curation, K.J., K.M. and T.S.; writing—original draft preparation, K.J.; writing—review and editing, K.J., T.S. and M.A.S.-M.; visualization, K.J.; supervision M.A.S.-M.; project administration, M.A.S.-M. and T.S.; funding acquisition, T.S. and M.A.S.-M.

**Funding:** This research was funded by a bilateral project of the Japan Society for the Promotion of Science (JSPS) and the Spanish National Research Council (CSIC).

**Acknowledgments:** This work was partly supported by the bilateral project of the Japan Society for the Promotion of Science (JSPS) and the Spanish National Research Council (CSIC).

**Conflicts of Interest:** The authors declare no conflict of interest.

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
