# Peer review of "The Efficiency of a Low Dose of Biochar in Enhancing the Aromaticity of Humic-Like Substance Extracted from Poultry Manure Compost"

_agronomy, doi:10.3390/agronomy9050248_

Reviewer 1 Report

Overall, the study is interesting and the findings can be very useful in different applications of compost. I would like to see a summary of the other research that has been done on compost and biochar by the researchers in this paper rather than just referring to as a reference.

Author Response

Specific response to referees

Remarks from Referee #1:

Overall, the study is interesting and the findings can be very useful in different applications of compost. I would like to see a summary of the other research that has been done on compost and biochar by the researchers in this paper rather than just referring to as a reference.

Our Answer: We would like to thank Referee#1 for suggestions. We acknowledge the comments. Taking into the account on the suggestion by Referee#1, additional sentences regarding the outcome of the previous research on biochar with compost have been inserted in the introduction. This section now describes the main results obtained in previous experiments, which justifies the current study.

Reviewer 2 Report

REVIEW

Summary

In this paper the authors aim at presenting the aspects of 2% hard-wood tree biochar addition on the chemical composition of humic-like substances in a composting process of poultry manure.

General comments

Recently, the application of biochar as a bulking agent in organic waste composting has become the focus of interest due to the numerous benefits obtained by the composting process, such as organic matter degradation and humification, reduction of heavy metals bioavailability, mitigation of NH3, N2O and CH4 emissions. Therefore, the paper addresses an important issue, but a constructive analysis of the published information is missing.

Furthermore, there are some articles with similar topic. A wide range of biochar application rates to compost have been tested (from 2% to 50%) with various (and similar) research objectives, including the influence on humification, so the manuscript should emphasize the novelty of this work.

The manuscript needs to be checked for grammar and syntax. There are a few sentences that require rephrasing for clarity.

I have a few additional comments and suggestions that may indicate points for further improvement. Please see the specific comments below.

Specific comments

·        ABSTRACT

o   "Abstract" is well written and reflects clearly the actual scope of this manuscript, but it’s too general and some important information is missing, such as biochar type and the duration of the composting process.

·        INTRODUCTION

o   Introduction is well arranged, but it seems a bit too general. It should be enriched with more detailed description of the beneficial and possible disadvantageous outcomes of biochar application in composting process. Synergisms between compost and biochar for soil improvement/amelioration is an important point concerning the MS.

o   Line 35: Please check the references. None of the given references addresses the effect of the combined application of biochar and earthworms.

o   Line 42-45: The author should complete this section. For example, the main results and consequences about OM degradation and humification due to the biochar application is only partly discussed in this part. I think, it would be an important point considering that this is an article about the biochar-mediated changes in aromaticity of humic-like substance.

 ·        MATERIALS AND METHODS

o   Description of the experimental part and methods is generally adequate, the applied monitoring methodology is correct. A lot of work is done to assess the efficiency of the treatments. However, some information is missing and some other is not clear.

§  Please give more details about the description of pyrolysis (e.g. residence time) and biochar properties. The authors gave the very similar description about these in cited article [25].

·        Keiji Jindo, Koki Suto, Kazuhiro Matsumoto, Carlos García, Tomonori Sonoki, Miguel A. Sanchez-Monedero, Chemical and biochemical characterisation of biochar-blended composts prepared from poultry manure, Bioresource Technology, Volume 110, 2012, Pages 396-404, ISSN 0960-8524, https://doi.org/10.1016/j.biortech.2012.01.120.

§  Line 67-68: Please give the applied methods for biochar characterization.

§  Line 67: ‘C = 791.5 g kg−1’. Please define ‘C’. Which method was used for the determination of ‘C’? Is it the same method as in 2.4. section?

o    Line 80: What is the reason that only one representative sample was obtained?

 ·        RESULTS AND DISCUSSION

o   Figures and Tables are informative, easily readable and correct.

o   Have been parallel measurements done from the sample? How was the pooled standard error of the chemical analysis determined? (Table 2)

There is not detailed information provided about significant differences (Table 2, Table 3)

o   Line 131: Please give more explanation about W2/W1 ratio and the relations to degree of humification.

o   Line 160: Table 3. - Please use bold letter.

o   Line 183. Please delete uppercase reference numbers in parentheses.

o   Line 193-196: This section contains only poor explanation and too general discussion on own results compared to the literature.

·        CONCLUSION

This part has to be improved with future directions.

·        REFERENCES

Line 286: Journal title abbreviation is not correct in this case; please check -https://www.ncbi.nlm.nih.gov/nlmcatalog?term=Biomass%20and%20bioenergy%5BTitle%5D

Author Response

Response to reviewer’s comments

We would like to thank Referee#2 for suggestions.

Our response for comments of the Anonymous Referee#2 was described as following:

Remarks from Referee#2:

Recently, the application of biochar as a bulking agent in organic waste composting has become the focus of interest due to the numerous benefits obtained by the composting process, such as organic matter degradation and humification, reduction of heavy metals bioavailability, mitigation of NH3, N2O and CH4 emissions. Therefore, the paper addresses an important issue, but a constructive analysis of the published information is missing.Furthermore, there are some articles with similar topic. A wide range of biochar application rates to compost have been tested (from 2% to 50%) with various (and similar) research objectives, including the influence on humification, so the manuscript should emphasize the novelty of this work. The manuscript needs to be checked for grammar and syntax. There are a few sentences that require rephrasing for clarity.

Answer:

We acknowledge the comments. The manuscript has been strongly improved by the feedback from this comment. Additional statistic analysis has been conducted to improve constructive analysis. In respect with the novelty of this work, in the revised version we have rephrased last part of the introduction section to clearly identify the main objective of this paper and its novelty, compared to the previous one. The improvement of English writing has been done in the revised version. We hope these changes successfully addresses the concern of the referee.

 Specific comments from Referee about Introduction:

o   Introduction is well arranged, but it seems a bit too general. It should be enriched with more detailed description of the beneficial and possible disadvantageous outcomes of biochar application in composting process. Synergisms between compost and biochar for soil improvement/amelioration is an important point concerning the MS.

Answer: A sentence related to synergisms between compost and biochar has been additionally inserted. Possible disadvantageous outcomes of biochar application in composting process are the cost for high amount of the biochar required which was already described in the previous version.

o   Line 35: Please check the references. None of the given references addresses the effect of the combined application of biochar and earthworms.

Answer: A new reference related to biochar wit earthworm has been inserted.

o   Line 42-45: The author should complete this section. For example, the main results and consequences about OM degradation and humification due to the biochar application is only partly discussed in this part. I think, it would be an important point considering that this is an article about the biochar-mediated changes in aromaticity of humic-like substance.

Answer: Additional sentence regarding possible mechanisms of the biochar interaction with humification has been inserted.

 Specific comments from Referee about Material & Method:

o   Description of the experimental part and methods is generally adequate, the applied monitoring methodology is correct. A lot of work is done to assess the efficiency of the treatments. However, some information is missing and some other is not clear. §  Please give more details about the description of pyrolysis (e.g. residence time) and biochar properties. The authors gave the very similar description about these in cited article [25].

Answer: Thanks for the comment. Some additional information was inserted in the revised version.

§  Line 67-68: Please give the applied methods for biochar characterization.

Answer: The applied methods are now described in the revised version

 §  Line 67: ‘C = 791.5 g kg−1’. Please define ‘C’. Which method was used for the determination of ‘C’? Is it the same method as in 2.4. section?

Answer: This analysis correspond to a total C content measured by elemental analysis. A brief description of the methodology has been included in the revised manuscript.

 Line 80: What is the reason that only one representative sample was obtained?

Answer: For the extraction of humic substances, it is generally preferable to get a large and representative sample of the whole composting pile (as described in the text). In this case, the selection of an appropriate and representative sample is more critical than the extraction procedure.

 Specific comments from Referee about Result & Discussion:

o   Figures and Tables are informative, easily readable and correct.

Answer: Thank you so much.

 o   Have been parallel measurements done from the sample? How was the pooled standard error of the chemical analysis determined? (Table 2)

Answer: It’s duplicated. The reference for the determination of the pooled standard error has been inserted in the revised version.

 There is not detailed information provided about significant differences (Table 2, Table 3)

Answer: The result of the statistic analysis for significant differences has been inserted in the revised version.

 o   Line 131: Please give more explanation about W2/W1 ratio and the relations to degree of humification.

Answer: The sentence was corrected. And the reference was added.

 o   Line 160: Table 3. - Please use bold letter.

Answer: Letter type was changed

 o   Line 183. Please delete uppercase reference numbers in parentheses.

Answer: Deleted.

 o   Line 193-196: This section contains only poor explanation and too general discussion on own results compared to the literature.

Answer: Additional sentences were inserted.

 Specific comments from Referee about Conclusion:

This part has to be improved with future directions.

Answer: A new sentence regarding future directions has been inserted.

 Specific comments from Referee about References:

Line 286: Journal title abbreviation is not correct in this case; please check

Answer: Corrected.

Reviewer 3 Report

 The study investigates the effects of biochar in composting on the properties of humified organic matter. The subject is of interest to the readers of Agronomy. A major concern I have pertains to the extraction procedure. I wonder how the authors ensured that extracted biochar/dissolved black carbon was not intermixed in their extracts and did not overlap all the measured signals. In addition, there are a few flaws that need to be addressed before the study can be published:

major

101 Ash content needs to be included in the mass balance to  determine O content from CHN measurements

129 please contextualize H:C and O:C molar ratios with literature values for better clarity

 Language editing is strongly advised as indicated by the following examples

102 delete “thermal”

114 followS

125 “proportion of” missing

134 “Biochar [..] contains a large surface oxygen functional groups”, do the authors mean “contains a high abundancy of oxygen functional groups”

179 “band” missing, band and “peak” are interchanged in but are not the same

183 delete (383940)

193 „been“ missing

Author Response

Remarks from Referee #2:

Remark from Referee #2:  The study investigates the effects of biochar in composting on the properties of humified organic matter. The subject is of interest to the readers of Agronomy. A major concern I have pertains to the extraction procedure. I wonder how the authors ensured that extracted biochar/dissolved black carbon was not intermixed in their extracts and did not overlap all the measured signals.

Our Answer:

We were so glad to receive the comment from Referee #2. This is, in fact, a key question regarding the interference of the presence of biochar, and we agree with the concern of the referee. This paper was indeed designed to overcome this problem observed in our previous paper where we studied the humification of the bulk organic matter (in this case the interference of biochar is evident). Concerning intervention of an extractable fraction derived from biochar during the alkaline extraction, we confirmed based on the study of the raw biochar that the amount released from biochar was negligible.

Additionally, the type of biochar used in our study was produced at 550 C, and the feedstock was hard-wood material. This type of biochar usually does not contain a high amount of volatile compounds which can be extractable by alkaline solution. However, it is possible that during degradation in the composting pile some fragments of biochar may be co-extracted and pass through the filters. We expect that if any biochar fraction, it would be precipitated with the HA, and not affecting FA. In our opinion, we expect to have only a minor disturbance of the biochar in the extracted fraction.

Remark from Referee #2: In addition, there are a few flaws that need to be addressed before the study can be published: 101 Ash content needs to be included in the mass balance to  determine O content from CHN measurements

Our Answer: We verified newly the methodology and the obtained data. And, we calculated newly each elemental content considering the ash content, as pointed by Referee #2. Table 2 includes now the elemental analysis of HA and FA calculated on ash-free basis. We appreciate this improvement.

 Remark from Referee #2: 129, please contextualize H:C and O:C molar ratios with literature values for better clarity

 Our Answer: We have included a few references where these indexes have been used to study HA and FA extracted from composting materials. Additional sentences were inserted to give more description on these indicators.

 Remark from Referee #2: Language editing is strongly advised as indicated by the following examples

102 delete “thermal”                                     Our Answer: Deleted.

114 followS                                                        Our Answer: “S” has been added.

125 “proportion of” missing                        Our Answer: Those words have been inserted.

134 “Biochar [..] contains a large surface oxygen functional groups”, do the authors mean “contains a high abundancy of oxygen functional groups”

Our Answer: Yes, indeed. We changed the sentence.

179 “band” missing, band and “peak” are interchanged in but are not the same

Our Answer: We agree. The sentence was changed.

183 delete (383940)                                        Our Answer: Deleted.

193 „been“ missing                                         Our Answer: Added.

Round  2

Reviewer 3 Report

The authors addressed all my comments appropriately

Author Response

Thank you for your comment.  We are glad that the new version has addressed properly your previous concerns. We acknowledge the comment. Newly, several sentences in the sections of Discussion and Conclusion have been inserted. Additional statistic analysis in Table 2 has been conducted to improve constructive analysis. Overall, the manuscript has been improved. We hope these changes successfully addresses the concern of the referee.